# Multiple Stimuli-Responsive Conformational Exchanges of Biphen[3]arene Macrocycle

**DOI:** 10.3390/molecules25245780

**Published:** 2020-12-08

**Authors:** Yiliang Wang, Liu-Pan Yang, Xiang Zhao, Lei Cui, Jian Li, Xueshun Jia, Jianhui Fang, Chunju Li

**Affiliations:** 1Center for Supramolecular Chemistry and Catalysis and Department of Chemistry, School of Materials Science and Engineering, Shanghai University, Shanghai 200444, China; steven93916@126.com (Y.W.); zhaoxiang_shu@163.com (X.Z.); cuilei15@usst.edu.cn (L.C.); lijian@shu.edu.cn (J.L.); xsjia@mail.shu.edu.cn (X.J.); jhfang@shu.edu.cn (J.F.); 2Academy for Advanced Interdisciplinary Studies, Southern University of Science and Technology, Xueyuan Blvd 1088, Shenzhen 518055, China; yanglp@sustech.edu.cn; 3Key Laboratory of Inorganic-Organic Hybrid Functional Material Chemistry, Ministry of Education, Tianjin Key Laboratory of Structure and Performance for Functional Molecules, College of Chemistry, Tianjin Normal University, Tianjin 300387, China

**Keywords:** macrocycle, conformational exchange, multiple stimuli-responsive

## Abstract

Conformational exchanges of synthetic macrocyclic acceptors are rather fast, which is rarely studied in the absence of guests. Here, we report multiple stimuli-responsive conformational exchanges between two preexisting conformations of 2,2′,4,4′-tetramethoxyl biphen[3]arene (MeBP3) macrocycle. Structures of these two conformations are both observed in solid state, and characterized by ^1^H NMR, ^13^C NMR and 2D NMR in solution. In particular, conformational exchanges can respond to solvents, temperatures, guest binding and acid/base addition. The current system may have a role to play in the construction of molecular switches and other stimuli-responsive systems.

## 1. Introduction

In nature, conformational exchanges are widely involved in various kinds of biological processes [1]. In chemistry, controlled conformational exchanges of single molecules present an opportunity to control the molecular world [2]. For example, overcrowded alkenes were designed and synthesized to construct molecular switches, and their conformations can be changed predictably in response to lights [3], redox processes [4,5], temperatures [6], etc. On the other hand, molecular switches can also be constructed by host-guest complexes [7,8,9,10]. For supramolecular receptors, controlled and large amplitude conformational interconversions have been observed upon binding a guest molecule [11,12,13,14,15,16,17] or an ion [18,19,20], but the reversibly conformational exchanges of macrocycles beyond their host-guest properties have been rarely discussed [21,22,23,24,25,26,27].

Synthetic macrocycles often have simple structures and fast conformational exchange kinetics [28], which are hard to detect by common kinetic techniques. Although conformational interconversions of macrocyclic acceptors can be observed in molecular recognition processes [29,30,31,32,33,34,35,36], slow exchanges of coexisting conformations of macrocycles have been rarely reported [37]. Developing slow conformational exchange systems of macrocycles can not only apply in the design and synthesis of molecular machines and devices [2,7,8,9,10] but also function as an important tool for host-guest binding mechanism analysis [37]. The assignment of the binding mechanism is often inconclusive in synthetic supramolecular systems because of the fast exchange kinetics [38,39]. Very recently, Jiang et al. demonstrated the strict analysis of the binding mechanism based on a macrocycle with slow conformational exchange [37].

Here, we report the stimuli-responsive conformational exchanges of 2,2’,4,4’-tetramethoxyl biphen[3]arene (MeBP3) [40,41,42,43,44,45,46]. The conformational exchanges between two conformers of MeBP3 were slow on the NMR timescale, and we successfully differentiated them in the ^1^H NMR and ^13^C NMR spectra of MeBP3. ^1^H-^13^C HMQC and HMBC spectra were employed to identify these two conformers and assign the 1D NMR signals. Particularly, stimuli-responsive conformational exchanges can be realized by solvent effect, temperature variation, guest binding and acid/base addition.

## 2. Results

### 2.1. Conformations Analysis

MeBP3 contains three 2,2′,4,4′-tetramethoxyl biphenyl sidewalls and three methylene linkers, which is structurally similar to those synthetic macrocyclic arenes like calix[*n*]arenes [28,30,47,48,49], resorc[4]arene [50], cyclotribenzylene [51], pillar[*n*]arenes [52,53,54] and other biphen[*n*]arenes [40,41,42,43,44,45,46]. Theoretically, there are two representative conformations for MeBP3, due to flipping of the biphenyl sidewalls (Figure 1a, MeBP3-I and MeBP3-II). For MeBP3-I, three biphenyl units are arranged in the same orientation, and flipping of one biphenyl panel of MeBP3-I can produce MeBP3-II [55]. Conformer MeBP3-I has D_3_ symmetry and thus should give rise to two singlets in ^1^H NMR for their aromatic signals. Conformer MeBP3-II has C_2_ symmetry and six singlets in the aromatic regions. If the exchanges between two conformers are slow on the NMR timescale, we can differentiate them in the NMR spectra of MeBP3.

In the ^1^H NMR spectrum of MeBP3 (2.0 mM), one set of two broad singlets and the other set of six broad singlets in the aromatic regions were observed (Figure 2a), indicating that MeBP3-I and MeBP3-II are coexisting in CDCl_3_. Two sets of broad singlets indicate that single bonds in biphenyl units rotate fast, but the overall exchanges between MeBP3-I and MeBP3-II were slow on the NMR timescale at room temperature. The ^13^C NMR spectrum of MeBP3 in CDCl_3_ was tested; six carbon signals for MeBP3-I and 18 carbon signals for MeBP3-II in the aromatic regions were observed (Figure 2b). ^1^H-^13^C HMQC and HMBC were employed to assign the 1D NMR spectra (Appendix A). The minor species was assigned to MeBP3-II, while the major species was assigned to MeBP3-I, and their percentages were calculated according to the integrals in ^1^H NMR spectrum, which are 73% for MeBP3-I and 27% for MeBP3-II at 25 °C in CDCl_3_. We examined the ^1^H NMR spectra of MeBP3 in CDCl_3_ at a high concentration of 100 mM (Appendix A). No obvious concentration effect was observed on the distribution of conformers.

We obtained two single-crystals of MeBP3 containing CH_3_CN molecules or *cis*-1,2-dichloroethene (*cis*-DCE) molecules in unit cells [39], namely, “CH_3_CN@MeBP3” and “*cis*-DCE@MeBP3” [46], respectively. The single-crystal X-ray diffraction data suggested that the structure of MeBP3 in CH_3_CN@MeBP3 could be assigned to MeBP3-I, while the structure of MeBP3 in *cis*-DCE@MeBP3 was MeBP3-II (Figure 1b). In the solid state, MeBP3-I exhibits a triangular topology, and three biphenyl sidewalls define a cavity suitable for the linear guests with 5.43 Å diameters [39]. Flipping one biphenyl sidewall in MeBP3-I to the other side can produce MeBP3-II, which was observed in *cis*-DCE@MeBP3.

### 2.2. Solvent and Temperature Effects in Conformational Exchanges

Molecular conformations are closely related to solvents and temperatures. We therefore tested the conformational exchanges of MeBP3 in different solvents and temperatures. MeBP3-I and MeBP3-II also coexist in CD_2_Cl_2_, acetone-*d*_6_, acetonitrile-*d*_3_, toluene-*d*_8_ and xylene-*d*_10_, but their percentages were different (for MeBP3-I is 87%, 88%, 86%, 85% and 80%, respectively) at room temperature (Figure 3b). However, in high polarity solvent such as DMSO-*d*_6_, no signals for MeBP3-II were observed (Appendix A).

Variable-temperature ^1^H NMR spectra were also examined in different solvents. At relatively high temperatures (≥50 °C), only signals for MeBP3-I were observed in all selected solvents (Figure 3c, Appendix A), suggesting that MeBP3-I is more thermodynamically stable than MeBP3-II. Decreasing the temperatures of MeBP3’s toluene-*d*_8_ solution from 100 °C to −60 °C led to gradual transformations of the chemical equilibrium (Figure 3c). For example, at 25 °C, the percentage of MeBP3-I was 85%, while the percentage of MeBP3-II was 61% at −60 °C. That is, in low polarity solvent of toluene-*d*_8_, the chemical equilibrium leaned toward the MeBP3-II at low temperatures. However, at high temperatures, MeBP3-I was the dominant species (Appendix A). In contrast, in mid-polarity solvents such as CDCl_3_, CD_2_Cl_2_, acetone-*d*_6_ and acetonitrile-*d*_3_, MeBP3-I was always the main conformer (Appendix A). One possible reason is that the temperature-triggered conformational exchange is a solvation-driven process, CDCl_3_, CD_2_Cl_2_, acetone-*d*_6_ and acetonitrile-*d*_3_ can solvate MeBP3, thereby freezing the changes of chemical equilibrium [21].

### 2.3. Guest-Binding and Acid/Base Addition in Conformational Exchanges

Suitable guests may efficiently switch the conformations of acceptors [29,30,31,32,33,34,35,36]. According to MeBP3’s cavity size (5.43 Å) [40] and π-electron rich characteristics [40,41,42,43,44,45,46], a linear secondary ammonium guest **1^+^** was chosen to study the binding properties of the macrocycle. As shown in Figure 4, in the ^1^H NMR spectrum of host-guest 1:1 mixtures, no signals for MeBP3-II were found, showing that the conformational exchanges were fixed in the presence of **1^+^**. Meanwhile, the proton signals of **1^+^** shifted upfield dramatically (for example, ∆σ = −0.40 ppm for the protons at 2.94 ppm), suggesting that an interpenetrated complex was formed. In other words, MeBP3-I can bind **1^+^** efficiently, and the host-guest association can fix the conformational exchanges. Furthermore, no signals for free guest and free MeBP3 were detected, suggesting that the guest exchange kinetics of complex **1^+^** ⸦ MeBP3 was fast on the NMR timescale. This is in contrast to the slow interconversion kinetics between two conformers.

NMR titration experiments were employed to measure the association constants (*K*_1_, *K*_2_) between two conformers of MeBP3 and **1^+^** [37]. When we gradually titrated **1^+^** into a CDCl_3_ solution of MeBP3 (1.0 mM, Appendix A), the obvious shifts were observed in both two conformers’ ^1^H NMR signals, showing that **1^+^** can be recognized by both MeBP3-I and MeBP3-II. Nonlinear curve-fitting (Appendix A) gave the association constants of *K*_1_ = (3.50 ± 0.54) × 10^3^ M^−1^ (**1^+^** ⸦ MeBP3-I) and *K*_2_ = (2.27 ± 0.18) × 10^3^ M^−1^ (**1^+^** ⸦ MeBP3-II), respectively, which further indicated that MeBP3-I can bind **1^+^** more efficiently.

As mentioned above, the host-guest association can fix the conformational exchanges at room temperature, and the macrocycle favored the conformer MeBP3-II at low temperature intoluene-*d*_8_. Next, we investigated the influence of guest-binding toward conformational exchanges in the toluene-*d*_8_ solution of MeBP3 at −60 °C. Upon addition of 1.2 equivalents (eq.) of **1^+^**, the ^1^H NMR (Figure 5b) spectra of MeBP3 suggested that the conformational exchanges were nearly fixed. In both instances the absence of resonances in the aromatic regions of ^1^H NMR signals suggested that the MeBP3-II nearly disappeared. Then, we expected that the **1^+^** can be deprotonated with the assistance of organic base *n*-Pr_3_N, which would dissociate the host-guest complex and resume the conformational exchanges. Upon further addition of 1.5 eq. *n*-Pr_3_N, the NMR spectrum almost recovered the resonances for a mixture of MeBP3-I and MeBP3-II (Figure 5b). After that, with the addition of 1.8 eq. CF_3_COOH, no ^1^H NMR signals for MeBP3-II were found in the mixture solution (Figure 5b). The re-protonation of dibutylamine re-associated the host-guest complex and fixed the conformational exchanges. That is, the conformational exchanges between MeBP3-I and MeBP3-II can be responsive to the guest binding and acid/base addition. This result also suggests that the exchange kinetics of host-guest complex is faster than the conformational exchange.

## 3. Discussion

In conclusion, we investigated the multiple stimuli-responsive conformational exchanges of 2,2′,4,4′-tetramethoxyl biphen[3]arene. Two preexisting conformers of MeBP3 were found in solutions with slow conformational interconversions. Structures of these two conformers were fully characterized by ^1^H-^13^C HMQC and HMBC NMR spectra. Besides, solvents, temperatures, guest binding and acid/base addition can dramatically influence the conformational exchanges. In relatively high temperatures, conformer MeBP3-I is the dominant species in all solutions, for example, percentage of MeBP3-I is >95% in CDCl_3_, acetone-*d*_6_, toluene-*d*_8_ and xylene-*d*_10_. In contrast, MeBP3-II is the main conformation in MeBP3’s toluene-*d*_8_ solution at −60 °C (61%). Secondary ammonium guest **1**^+^ can enter the cavity of MeBP3 and fix the conformational exchanges, even at −60 °C in toluene-*d*_8_, whereas only MeBP3-I was observed in the host-guest mixture’s solution. The addition of acid/base can resume/re-fix the conformational interconversions of MeBP3 in the host-guest mixture’s solution. This slow conformation-exchange system may have a role to play in the binding mechanism analysis, molecular switches and stimuli-responsive materials. Studies directed to exploring these possibilities are underway.

## 4. Materials and Methods

2,2′,4,4′-tetramethoxy biphen[3]arene (MeBP3) was synthesized according to our previous report [39]. Secondary ammonium guest **1^+^**·BArF^−^ was synthesized according to the literature [29]. ^1^H NMR (acquisition time = 3.17 s, relaxation delay = 1.00 s) and ^13^C NMR (acquisition time = 1.09 s, relaxation delay = 2.00 s) spectra were recorded on a Bruker AV500 instrument (Bruker Daltonics Inc., Rheinstetten, Germany). ^1^H-^13^C HMQC spectrum, ^1^H-^13^C HMBC spectrum and 2D NOESY NMR spectrum were recorded on a Bruker AVANCE III HD 600 MHz spectrometer (Bruker Daltonics Inc., Rheinstetten, Germany). Variable-temperature ^1^H NMR (acquisition time = 2.19 s, relaxation delay = 5.00 s) spectra were recorded on a JEOL JNM-ECZ400SL NMR spectrometer (JEOL, Akishima-shi, Japan).

## Figures and Tables

**Figure 1 molecules-25-05780-f001:**
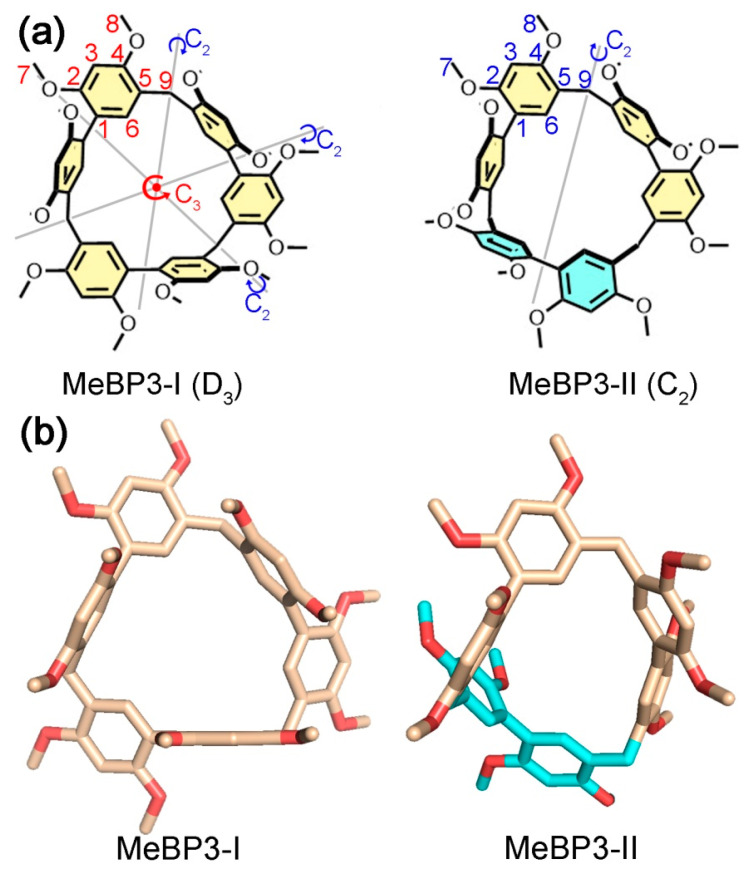
(**a**) Chemical structures of the two conformers of MeBP3. (**b**) Single-crystal structures of two conformers of MeBP3.

**Figure 2 molecules-25-05780-f002:**
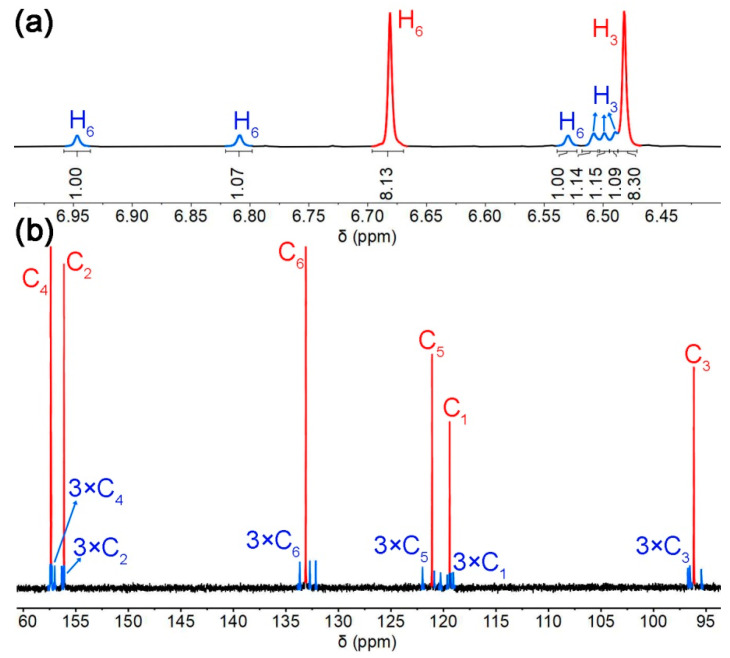
(**a**) Partial ^1^H NMR spectrum (500 MHz, 25 °C, 2.0 mM) of MeBP3 in CDCl_3_; (**b**) Partial ^13^C NMR spectrum (125 MHz, 25 °C, 100 mM) of MeBP3 in CDCl_3_.

**Figure 3 molecules-25-05780-f003:**
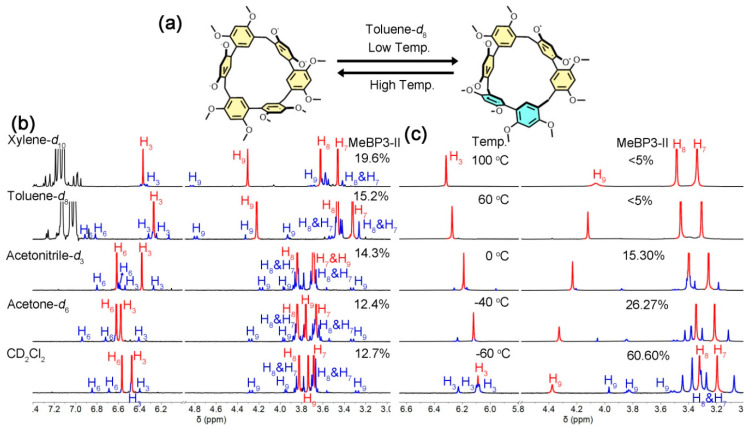
(**a**) Conformational exchanges of MeBP3. (**b**) Partial ^1^H NMR spectra (500 MHz, 25 °C, 2.0 mM) of MeBP3 in different deuterated solvents. (**c**) Variable-temperature ^1^H NMR spectra of MeBP3 in toluene-*d*_8_ (400 MHz, 2.0 mM).

**Figure 4 molecules-25-05780-f004:**
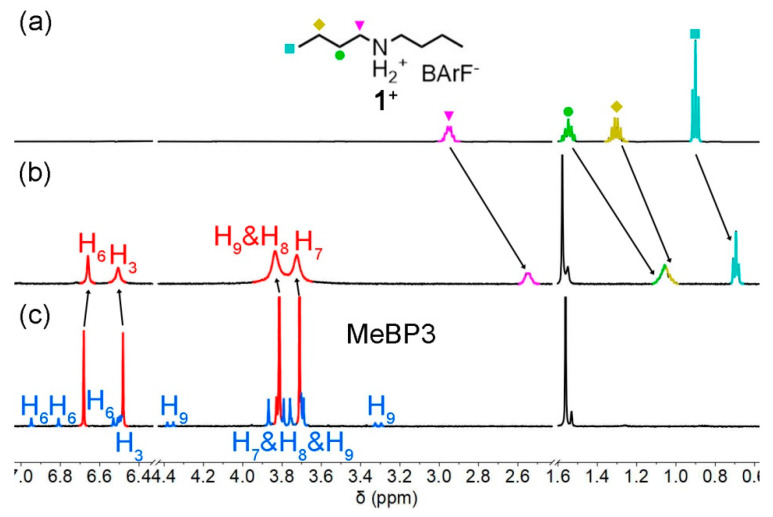
Partial ^1^H NMR spectra (500 MHz, 25 °C, 2.0 mM) of **1^+^** (**a**), host-guest 1:1 mixtures (**b**), and MeBP3 (**c**) in CDCl_3_.

**Figure 5 molecules-25-05780-f005:**
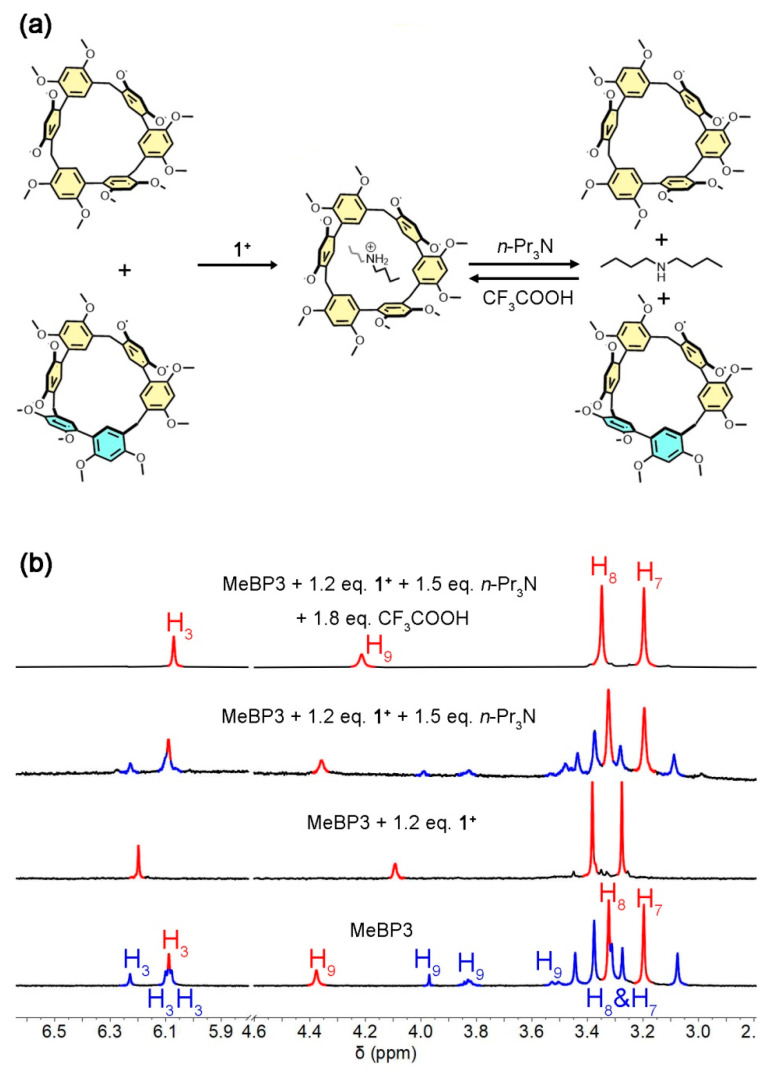
(**a**) Influence of guest-binding and acid/base addition toward conformational exchanges. (**b**) Partial ^1^H NMR spectra (400 MHz, −60 °C, 2.0 mM) of MeBP3 in toluene-*d*_8_ solution with different conditions.

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
