# Peer review of "Multiple Stimuli-Responsive Conformational Exchanges of Biphen[3]arene Macrocycle"

_molecules, 2020, doi:10.3390/molecules25245780_

Round 1

Reviewer 1 Report

The manuscript "Multiple Stimuli-responsive Conformational Exchanges of Biphen[3]arene Macrocycle" by Li and co-workers describes the study of conformational exchanges of 2,2’,4,4’-tetramethoxyl biphen[3]arene (MeBP3) under various conditions. The authors showed by 1H NMR, 13C NMR and 2D NMR spectroscopy that conformational exchanges are response to solvents, temperatures, guest binding and acid/base addition.

The manuscript is well written. I had a favourable impression when reviewing this manuscript.

However, I have questions and comments on the paper.

1) There are a number of relevant publications on stimulus- responsive (guest addition, acid/base addition) conformational exchanges in pillar[n]arenes that the authors do not provide in the introduction. Such examples should be added.

2) An explanation of the difference between toluene-d8 and all other deuterated solvents should be added to the text of the manuscript. Is this related to possible "guest-host" binding of the toluene molecule? It is also unclear why there is practically the same effect of influencing conformational changes in solvents with different nature and polarity (CDCl3, CD2Cl2, acetone-d6, acetonitrile-d3 and p-xylene-d10).

3) Why did the authors choose di-n-butylamine as a guest? An explanation for this choice needs to be added.

4) Why didn't the authors use 2D NOESY NMR spectroscopy?

5) The caption to Figure 5 does not match the image.

Reviewer 2 Report

This manuscript describes conformational changes in a macrocyclic structure upon different conditions. However, the text is poorly presented, while the overall work scope and experimental novelty are very limited so as to be considered this an original article. Moreover, the results are portrayed without a clear purpose, thus meaning that a conclusion (or at least an implicit one) cannot be subsequently extracted. Finally, the whole work presents serious language limitations which in turn yield an imprecise interpretation of the described results. Therefore, this manuscript cannot be accepted for publication in Molecules.

Reviewer 3 Report

The authors show how 2 conformers of their macrocycle coexist in solution and how the distribution varies depending on the solvent, temperature, and inclusion of a guest. The studies are well executed and described clearly. I would recommend the publication of this manuscript in "Molecules", provided that the following points get addressed:

1) The authors describe the trends they observed in different solvent and temperatures, but these results are not rationalised: can the authors explain the trends observed?  An attempt to do so would make the manuscript more interesting.

2) In the Discussion the authors comment that "this slow conformation-exchange system may have a role to play in binding mechanism analysis". Can the authors explain how they envision this?

3) The caption of Figure 5 indicates that the NMR spectra are measured in CDCl3 at 25C, while the text indicates that these were measured in toluene at -60C. I guess that it is the caption that is wrong and should be corrected.

4) The concentrations at which the NMR spectra were recorded should be indicated. Have the authors observed any effect of the concentration of MeBP3 on the distribution of conformers? 

5) The NMR parameters should be provided, especially the acquisition time and relaxation delay. Have they determined T1 values and verified that the relaxation delay is sufficient for quantitative NMR?

6) The authors give the distribution of conformers as obtained from the 1H NMR spectra in percentages with 1 (in the text) or even 2 decimal places (in Figure 3). In the SI the authors seem well aware that 1H NMR spectroscopy has a detection limit. This limit in accuracy of the integrals is also evident in Figure 2a, where different H signals of ME-BP3-II show a variation of up to 15%. I would strongly suggest to round all values to full%, unless the authors can proof that their error bar is equal to or smaller than 0.1%.

7) The English of the manuscript should be checked carefully, as it contains numerous errors. A few examples are:

  • are coexisted -> are coexisting
  • in the crystalline -> ???
  • in the present of -> in the presence of
  • the dominated species -> the dominant species
  • "Relative researches are underway" ->???

Round 2

Reviewer 1 Report

All the necessary changes have been made.

Reviewer 2 Report

The Authos have improved the manuscript and I now find it suitable for publication in Molecules